# The Relationship between Emotional Intelligence and Educators’ Performance in Higher Education Sector

**DOI:** 10.3390/bs12120511

**Published:** 2022-12-15

**Authors:** Osama Khassawneh, Tamara Mohammad, Rabeb Ben-Abdallah, Suzan Alabidi

**Affiliations:** 1Lazaridis School of Business and Economics, Wilfrid Laurier University, Waterloo, ON N2L 3C5, Canada; 2College of Business Administration, American University in the Emirates, Dubai 503000, United Arab Emirates; 3College of Education, Al Ain University, Abu Dhabi 112612, United Arab Emirates

**Keywords:** competencies, emotional intelligence, higher education, performance effectiveness, knowledge, structural equation modelling

## Abstract

The significance of emotions in the classroom has been thoroughly explored, but discussions on educators’ abilities to recognize, regulate, and manage their emotions are still ongoing. This paper aims to look at the concept of emotional intelligence (EI) and how professors in higher education can use it to achieve better results in the form of emotional intelligence competencies (EIC). A total of 312 educators from 25 higher education institutes in the United Arab Emirates (UAE) participated in this study. In sampling the Emotional Intelligence Competencies for this study, we adopted Costa and Faria’s (2015) EQ test, administered to the respondent. The Reuven Bar-On emotional intelligence scale was created and standardized to gather data. Using structural equation modeling, the validity and utility of a proposed model for EI-based teaching competencies and their relationship to critical strengths were evaluated (SEM). The findings show that EIC significantly impacts educator behavior, which in turn improves student success. In order to ensure successful instruction and remarkable performance, the study provides valuable recommendations to higher education institutes about the importance of recruiting new instructors with high skills in EI and providing training sessions for existing educators to improve their EI skills.

## 1. Introduction

Higher education in the United Arab Emirates (UAE) has expanded rapidly over the past decade, and the country hopes to become a regional hub for learning. The National Strategy for Higher Education 2030 was unveiled recently. This plan aims to ensure that the next generation of Emiratis can access the best possible scientific and professional training [1]. Educators play a vital role in the strategy as a catalyst for change that will help the UAE achieve its objectives [2,3]. Educators face obstacles in their course work, including accreditations, class management, a hectic classroom setting, insensitive management, and high expectations from students’ parents [4]. Instructors experience mental anguish, discontent, and emotional outbursts or fallout in such settings [5], and some even opt for early retirement as a result [6]. Consequently, being an educator is becoming more challenging and multifaceted than ever [7].

Similarly, [8] revealed that educators are among the professionals with the highest level of job stress, based on studies conducted across different cultures. To boost student achievement, some scholars and authors in the field have recently begun investigating the function of emotions and EI in the classroom [9]. The first study examined the positive effects of EI on the teaching profession [10]. EI is thought to be a predictor of things like job satisfaction and job performance [11,12]. Educators adept at recognizing and appreciating their own emotions are in a better position to express what they require and take the steps necessary to meet their emotional needs, which leads to achieving their professional goals [13]. They are more likely to be sensitive to the feelings of others and offer encouragement in order to inspire those around them to work together productively toward a shared goal. According to the results of these studies, educators who score highly on emotional intelligence have more ability to influence others [8,14]. The teaching profession is in crisis; more and more educators are deemed unprofessional and demoralized [15]. With that in mind, this research aims to illuminate the connection between EI and teacher effectiveness.

## 2. Theoretical Foundation and Hypotheses Formulation

In recent years, educators have come to value the role of emotions as a critical component of effective lesson planning and delivery [16]. Due to the large number of interpersonal interactions required of them, teachers must be able to control their own emotions and those of their students, parents, and coworkers [17]. Therefore, teaching is considered one of the most stressful careers because teachers’ emotional management affects their students’ learning, well-being, quality of social interactions, and academic achievement [18]. Today’s students have higher social, emotional, and technological literacy, so their educators need to constantly evolve and improve their EI to meet their students’ needs. Exceptionally high levels of professional stress are placed on college and university faculty. Interactions with students will elicit many positive and negative feelings [19,20]. According to [21] research, stress can result in either positive or negative emotions. 

Positive emotions in the classroom have increased students’ flexibility and created a more conducive learning environment [14,22]. Most negative feelings stem from students and teachers being at odds, which is also the primary source of stress, exhaustion, and attrition in the teaching profession. Several factors can contribute to such differences of opinion [23]. Teachers can disagree with anyone, including the administration, coworkers, and even the parents of their students [10,24]. Educators who maintain a healthy balance of positive and negative emotions can better help their students achieve their full potential. Therefore, self-regulation is a crucial skill for any educator working in higher learning. Teachers’ emotional regulation emerges from their knowledge, beliefs, attitudes, and emotions, as stated by [25], making it a necessary functional component for effective instruction. Mayer and Salovey first used the term EI to describe a person’s ability to perceive emotions, integrate emotion to facilitate thought, understand emotions, and regulate emotions to promote personal growth [26].

For this purpose, the EI is widely regarded as an essential component in raising educators’ overall effectiveness and to regulate the emotions of the educator. It could improve individuals’ skills, knowledge, attitude, as well as their capacity for self-reflection and self-management. According to [27], an emotionally intelligent teacher can control his negative emotions, maintain a positive outlook, and motivate himself and his students. Several studies on teacher efficacy have examined the link between emotional intelligence and how people deal with conflicts. According to the findings of this study, emotionally intelligent educators can better manage their interactions with students by favouring the cooperation type in addition to the avoidance style [14,28,29,30]. Even though emotions and emotional intelligence (EI) are integral to education, research on their respective roles in the classroom has been scant [31]. 

Researchers know surprisingly little about the role of emotions in learning to teach, how teachers’ emotional experiences relate to their teaching practices, and how the sociocultural context of teaching interacts with teachers’ emotions [32]. Educators’ feelings and how they deal with them should be looked at in terms of the most essential parts of their jobs, focusing on how well teachers understand themselves [33,34]. EI must be recognized in professional teaching standards, and its significance in developing capable and experienced educators must be understood [35]. One of the more recent pieces of literature on this subject is [13] study, which emphasizes the importance of emotional intelligence (EI) for college professors. It encapsulates the central idea that educators are responsible for assisting their students in learning and, as such, must be aware of the importance of students’ feelings throughout the classroom. To make it work, they will need to rely on their EI. The third and most important factor is one’s emotional quotient (EQ). An excellent educator assists students in developing emotional intelligence by providing them with knowledge, expertise, and effective learning strategies. In the classroom, emotional intelligence (EI) would entail self-awareness and empathy for the students, focusing on creating an environment conducive to learning for all [36]. It is still unclear how teachers become self-aware of their emotional states in the workplace, how they manage those states, and how they use that knowledge to exercise emotional agency [9,12,37].

Conversations with students about their expectations, attentive listening to what they have to say, growing self-awareness, and other similar activities are some behavioural indicators of emotional intelligence in the classroom. A large part of this can be understood if the previously mentioned behavioural indicators are expressed in terms of emotional intelligence competencies (EIC). Some researchers believe EIC is a significant factor in determining a teacher’s instructional efficacy [15]. EIC is critical not only for academic development but also for students’ psychological and interpersonal development [38]. In addition to helping students, EIC can help educators grow by encouraging them to keep a positive attitude, find meaning in their work, and inspire those around them. It stands to reason that a happy, fulfilled educator would be more effective in the classroom than a dissatisfied one. EI affects teachers’ skills because it affects how they think and act in many different ways. Several authors link emotional intelligence (EI) to various pedagogical skills and characteristics [39]. Table 1 shows the relationships between these concepts.

According to personality research, emerging adulthood is associated with a period of “maturity” characterized by increased conscientiousness and emotional stability [40]. Most longitudinal research in emerging adulthood has focused on the predictive utility of the first one or two years of life regarding trait emotional intelligence (EI). However, researchers have paid scant attention to how the structure evolves over the long term [41]. Furthermore, analyzing the construct’s stability is critical because it can provide insight into how malleable or responsive the construct is to interventions during this period [42] say that common statistical techniques for gauging the stability of a construct over time include rank-order stability and average change.

Critically reviewing the cited literature reveals that EI influences a teacher’s competencies to achieve higher performance levels and improve teaching quality. However, the proposition’s validity requires a numerical demonstration of these observations [55].

This gap in the existing literature inspired the topic of this study. The readers of this study will be introduced to the idea of emotional intelligence (EI) in educators, and they will be shown how EI can be integrated into effective teaching as EIC in higher education instructors. This article introduces readers to the concept of emotional intelligence (EI) in academics.

Therefore, the following hypotheses was formed: There is a positive relationship between emotional intelligence training and educators’ performance. There is superior performance when emotional intelligence competencies are incorporated in higher education institutions.

It leads to developing the initial model (shown in Figure 1) that represents the interplay between the various skill sets. A model depicting the connection between EI and pedagogical abilities has been developed based on the extensive literature review described above.

## 3. Research Method

To gain a deeper understanding and a holistic picture of the social world, a quantitative approach was utilized in this study to determine the impact of EI in higher education sector in the UAE. Furthermore, this quantitative research produces factual data that can be clearly explained through statistics [56].

A literature review uncovered the EIC, and a theoretical model was developed. Structural equation modelling (SEM) was used to analyze the data and verify the model. Numerous procedures have been described in this section. The EIC for this study was adapted from the EQ test developed by [57] and administered to the respondents in the ways detailed below. We zeroed in on this test because it applies to the UAE setting and has been tried and tested in the education sector. Individuals are evaluated based on their emotional Intelligence (EI), as defined by [58], which is the capacity to appropriately and successfully respond to various emotional signals arising from one’s internal experiences and those in one’s immediate environment. Sub-indicators for emotional sensitivity, emotional maturity, and emotional competency are included in the assessment. In addition, it specifies the various dimensions and outward signs of conduct that can be utilized to foretell an individual’s EI. Emotional intelligence and emotional competency are two distinct but interconnected abilities. In this study, we refer to the competencies found in an EQ test’s maturity and sensitivity sections as EIC.

The EQ test, on the other hand, says that “emotional competency” is a small set of skills that can be measured by actions [33]. Information was gathered using a standardized version of Reuven Bar’s emotional intelligence measure. Users can learn more about their emotional and social capabilities using the emotional intelligence scale. After taking EQ, one can look at a report that breaks down their results by skill area [40]. The following emotional intelligence scale was chosen because of its widespread use as a diagnostic tool.

Other crucial aspects of EI, such as emotional awareness, self-control, and practical application, are also evaluated. In its research methodology, this work takes a reductionist stance. Ref. [59] explains that the deductive method entails developing a theory-based hypothesis (or hypotheses) and then designing an empirical investigation plan to test the hypothesis. One way to account for deductive thinking is through speculation, which can be derived from the theory’s premises. The deductive method aims to conclude from premises or propositions that have already been given. Ref. [12] claims that deductive reasoning has the potential to explain causal linkages between concepts and variables, the potential to quantify concepts quantitatively, and the potential to expand research results to some extent. In the publication, the authors provide two testable hypotheses. Data from the respondents have provided empirical validation of the model. Three hundred and sixty-six professors from twenty-five universities in the United Arab Emirates (UAE) were contacted electronically (five public and twenty private). As a result, 312 responses were collected. Table 2 shows our variable description. 

The questionnaire contained a section on self-evaluation and a sub-section on emotional quotient. Respondent educators come from all levels of academia and range in age from 25 to over 70. A modified version of Costa and Faria’s Emotional Quotient Test (EQ) was administered to the teachers in this study (2015). As for the evaluation’s quality, it scores 0.88 on validity, 0.93 on test–retest reliability, and 0.88 on split-half reliability. In addition to the EQ test, educators were asked to rate themselves on various other indicators, such as classroom impact, administrative duties, scholarly output, and student evaluations. A Likert scale from 1 to 5 was employed for this method of self-evaluation, with one being the lowest and five the highest. Using a statistical technique called structural equation modelling (SEM), researchers can check for connections between observable and unobservable (construct or latent) variables. As with multiple regression equations, it examines the structure of relationships expressed in a series of equations to shed light on the interplay of multiple factors. Multiple regression and factor analysis form its analytical backbone. Three primary features define SEM. To begin, there is the task of determining intricately intertwined dependencies. Second, having the capacity to represent ideas that are not immediately apparent in these exchanges. The third step is constructing a model that accounts for all the interconnections. Smart PLS software was used to analyze the data via the partial least squares (PLS) method of SEM for path modeling. When only a small amount of information about the latent variables is known, the variance-based PLS method is preferred over the covariance-based SEM method for assessing descriptive and predictive correlations [60,61,62].

Additionally, PLS-SEM is favoured when theory creation is the primary focus of an exploratory study [62]. Since this research proposes a new theory about teacher abilities, it is appropriate to employ the PLS method. Sample sizes in the inner route model should be at least ten times larger than the most significant number of structural routes leading to a given construct. Standard SEM models consist of two models [14,62]. The first model, known as the measurement model, illustrates the combination of measured variables that yields a construct, while the second model, known as the structural model, demonstrates the interrelationships among constructs. As the measurement model is developed, it must account for the construct’s observable components. The EQ assessment reveals a three-dimensional structure for the EIC’s constituent variables. Emotional intelligence can be broken down into three categories: maturity, stability, and skill [31]. We retrieved the observable components of knowledge, skill, attitude, and performance from the study by [55], who classified the essential competencies for effective teaching as knowledge, skill, and attitude. Content knowledge (CK), practical knowledge (PK), and a dedication to lifelong learning can all be used to gauge a student’s development as they work toward their goals (CL). The ability to analyze and instruct (AIS), communicate (CS), manage a classroom (CM), plan and organize (POS), and network (NS) are all quantifiable skills. Attitude can be measured in several ways, including open and adaptive attitude (OAA), open and optimistic attitude (EOA), and positive attitude (PS) (OA). Performance is evaluated along four dimensions: self-efficacy (SE), administrative duties (ADM), student feedback (SF), and innovation (R&D). As indicated before, the structural model is the first proposed model depicting the connection between EIC and knowledge, skills, and attitude (Figure 1). According to the authors, EIC positively impacts a teacher’s fundamental competencies, which leads to higher output. The data was analyzed using the Smart PLS program.

## 4. Results and Discussion

Approximately 39% of the educators surveyed fell into the 25–30 age range, 13.2% were between the ages of 31–40 and 29.6% were between 41–50 age range, 13.2% were between the ages of 51–60, and 5% were 60 and above. The study included 43% male participants and 57% female participants. Assistant Professor made up the majority of respondents (63%), followed by associate professors (17%), professors (9%), and lecturers (5%). Table 3 provides more details about the demographic characteristics of our sample.

Educators’ Emotional Intelligence (EI) was broken down into three categories based on their EQ test scores: emotional receptivity, emotional maturity, and emotional competence. Descriptive statistics indicate that instructors have a very high mean EI of 368/440 based on the results of the EQ test. Scores for sensitivity (92/100), emotional maturity (117/140), and emotional competence (165/200) were all very high (extremely high). The results suggest that most respondents had above-average levels of emotional competence, such as self-control, ego-management, and dealing with emotional upheavals. While sensitive and mature, this seemed lacking in comparison.

Teachers still need to be more student-centered, as maturity includes self-awareness and concern for the growth of others (students). They need to focus more on their students’ growth as whole people and modify their methods of instruction accordingly to meet the needs of pupils with different proficiency levels. Sensitive people have empathy, better relationships with others, and the ability to express their feelings clearly [43].

Teachers, then, need to have an excellent emotional meter and be willing to acknowledge students’ feelings and perspectives. Teachers should build rapport with their students so that they feel comfortable coming to them with issues. Instructors should also encourage students to feel and express joy and gratitude. Mature behaviour was also found to be most common in people aged 60 and up, particularly in academics. This is because, as a teacher gains wisdom and experience, he can better evaluate his performance. He may have reached all his career goals, overcome all his obstacles, big and small, and settled into a reasonably solid routine. As the rage of adolescence subsides, he develops a more levelheaded and composed demeanor, making him more malleable and pliable. It also frees his attention to concentrate on his students and contribute to their overall growth [60,63,64].

Furthermore, the highest sensitivity score was seen in this age range, suggesting that more seasoned educators have a deeper grasp of their students and are better able to connect with them. This is because they spread joy and encourage students to participate in class discussions. There was a marked improvement in emotional competence across the board, except for the middle-aged. New challenges, responsibilities, and the desire for familiarity are usually to blame. 

Nowadays, professionals must consider their students, jobs, and families. Because they are just starting their academic careers, younger age groups and assistant professors tend to have less emotional control because they have not yet learned how to handle their egos and control their feelings. When the data was broken down by gender, it was found that women performed best in sensitivity but worst in incompetence. The results corroborate to previous studies and hypotheses suggesting that females are more sentimental and less pragmatic than males [65,66]. The upshot is that those female educators have a leg up when connecting with and supporting female students. 

Data were analyzed with Smart PLS software using the partial least squares (PLS) method within SEM for path modeling. To begin with, we have the measurement model, which demonstrates how constructs are generated from measurable variables. The structural model, in contrast, reveals the interconnections between these concepts. Composite reliability (CR) and Cronbach’s alpha for reliability, AVE (average variance extracted), and factor loadings for convergent validity and discriminant validity are all ways to assess the measurement’s (or the outer model’s) fitness [67]. Table 4 displays the results of the estimated CR for the reliability test; all values are above the cutoff of 0.8. (Cheah et al., 2018). See Table 4. 

Most scales were close to the ideal value of 0.50 when convergent validity was measured using the AVE criterion developed by [68]. Looking at how much weight each variable has in describing the construct is another way to test for convergent validity (Table 5). Every factor loading has a statistically significant value close to or greater than the target value of 0.8. There has been some research on this topic [69,70]. The AVE test and the correlations established by Schoenherr et al. were used to evaluate the AVE’s discriminant validity. The R-squared, path coefficients, f-squared, and Q-square Stone-Geisser tests are all ways to evaluate the validity of an internal or structural model [7,67]. R-squared values are displayed in Table 5. A strong R2 value is 0.65, a moderate value is 0.35, and a weak value is 0.18 [71]. The proper t-statistics were determined with the help of bootstrapping, as well. Once the corresponding T-statistics for these parameters are larger than 1.96 (95% confidence level), we can say that they are statistically significant. Only the emotional intelligence (EI) to performance (performance) and the knowledge (knowledge) to performance (performance) instances had values below 1.96.

It is clear from this that neither EI nor knowledge has any bearing on performance. More than possessing extensive knowledge is required for success as a teacher. This supports our original claim that knowledge without practical abilities and mindsets will not make a teacher more effective. The variance inflation factors (VIF) for all the constructs are under the three-point limit recommended by the social science literature, so multicollinearity is not a major problem [72]. The route coefficients determine the impact of one variable on another. The SEM model results for attitude competencies reveal that EI has a substantial but insignificant impact (0.882). Although we discovered no statistically significant link between EI and performance, we did find a total effect (including indirect links) of 0.322. Acquiring new information affects one’s ability to do things (0.923). Attitude-related qualities are the most critical factors in successful performance (0.638). Skills have a modest effect on performance, as indicated by the path coefficient of 0.409. When the role of knowledge in improving performance is isolated, it is found to be relatively minor (0.129). A statistically significant relationship exists between increased knowledge and performance (0.590). This may suggest that all teachers have roughly the same level of knowledge due to the standard requirements for becoming teachers. More effort must be made to ensure that information is transmitted effectively. Knowledge alone does not make a good teacher; successful knowledge transmission demands additional skills and attitudes, which an individual’s EQ strongly influences. Hypothesis 1 of the paper is supported by the data. Performance can be affected by emotional intelligence. Emotional intelligence is essential, but it needs to boost productivity by itself. Skills and knowledge base influence educators’ emotional intelligence (EI) and motivation to succeed.

Culture plays a role in how people internalize and articulate their feelings [73]. Communicating feelings nonverbally is one example of expressive control that one’s cultural background could influence. Educators who spend extended periods working and living abroad often develop a new way of expressing their feelings [74]. To better integrate into their host culture, ex-pat educators’ emotional patterns take on the traits of their local peers rather than those of their home country’s peers. Their cultural backgrounds may shape people’s self-perceptions of their emotional intelligence and outward displays of emotion. Self-reported emotional Intelligence (EI) may, for instance, be influenced by social norms regarding what kinds of emotions are deemed appropriate [59,75]. An extraordinarily expressive and extroverted person would rate themselves lower in expressive control if raised in a culture that valued emotional repression and a calm demeanor. Since their personality traits align so well with display guidelines and cultural norms, extroverted people in a nation that places a premium on emotional expression can consider themselves to have excellent expressive control [76,77,78].

## 5. Conclusions

The research indicates that emotional intelligence (EQ) is connected to almost every aspect of working in the academic field. A successful educator must possess various necessary qualities, including knowledge, skills, and a positive attitude; however, more than these qualities are required to ensure success. Nevertheless, something else still plays a part in determining the level of efficacy with which these abilities are displayed. The sum of all of these factors determines efficacy. The EI competencies help bridge this gap by impacting a teacher’s knowledge, abilities, and perspective, which, in the end, results in an effective educator. The approach provides empirical backing for the idea that EI is critical in education and that educators should have adequate EIC to improve student outcomes. This idea is that EI is critical in education and that educators should have adequate EIC. This notion is that EI is essential in education and that educators should have sufficient access to it. There is an overwhelming majority of academic support for providing teachers with training to assist in the growth of their EI. To put it another way, a teacher with a higher EIC will be more effective at communicating his knowledge, determining the needs of his students, demonstrating love to them, and gaining their trust. Additionally, he or she will be better at gaining the students’ confidence. They can propel their own development to new levels if they work on strengthening their connections with other people. This suggests that he could benefit from EI not only in his interactions with students but also in his growth as a person, and it is not just limited to his interactions with students.

Therefore, educational institutions should invest in training programs that focus on emotional intelligence to cultivate emotionally intelligent educators. This will allow the institutions to serve their students better. Because teachers are a school’s most valuable resource, the administration should make encouraging their ongoing professional development one of their top priorities. This is because teachers are the school’s most valuable resource. EI is not a nice-to-have but rather an essential part of a teacher’s job, so they should give it the same amount of care and attention as they do to their subject matter and instructional methods. Since emotional intelligence is not a nice thing but rather an essential part of a teacher’s job, it is difficult to draw broad conclusions about emotional intelligence from this study due to its limitations, specifically the tiny sample size. Further research must be conducted, with the primary emphasis being on this topic. In addition, the research process is hampered by the deductivist school of philosophical thought. It is necessary that every one of the premises produced by the inductive research be accurate and that the words be specified appropriately for the conclusions reached through the process of deductive reasoning to be valid. When conducting research in the future, it will be necessary to examine several distinct philosophies from various perspectives side by side to reach a consensus regarding the approach that will be most successful.

## Figures and Tables

**Figure 1 behavsci-12-00511-f001:**
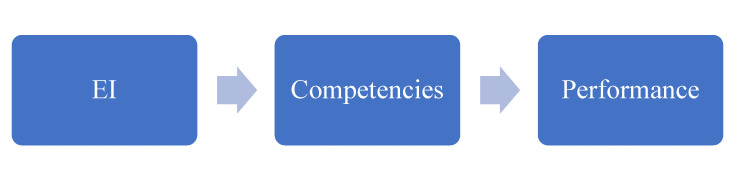
The Proposed Model. Source: The authors.

**Table 1 behavsci-12-00511-t001:** Teaching Characteristics Influenced by EI.

Teaching Characteristics Influenced	Authors
motivation, attribution, efficacy, beliefs, and objectives come from within.	[36,13]
maintain healthy relationships	[28,4,43]
understand students	[5]
striving to achieve favourable targets, flexibility to adopt	[10]
more concerned, involved, and adaptable to the needs of students,	[44,45]
motivation	[46]
devise proactive strategies for coping with stress and work pressure, job satisfaction, and motivation	[29]
positive environment, optimal learning and healthy development	[47]
facilitates knowledge delivery and exchange	[48]
participative and eager environment	[49]
empathy and caring disposition	[50]
environment of trust	[51]
the environment of overall growth	[20]
creates a learning classroom environment	[52]
strategies for resolving conflicts with teachers, classmates, and parents	[53]
ability to lead, make decisions, interact with others, collaborate, think creatively, and gain parents’ confidence in the classroom.	[54]

**Table 2 behavsci-12-00511-t002:** Variable description.

EI	Emotional intelligence refers to a person’s awareness of, and mastery over, their emotional state. This allows them to effectively manage their emotions in social and professional settings, lowering stress levels, improving communication, increasing empathy, overcoming challenges, and defusing conflict.
Attitude	The emotional state that affects one’s behavior. One’s feelings towards one’s work, employer, or teammates can affect one’s evaluation of those entities. Similarly, one’s mood might affect their actions at work. Researchers have found that close friends and family have the most significant emotional Intelligence.
Knowledge	Information is created by EI, where a person accesses and makes meaning of their feelings, and knowledge can be developed.
Performance	Teachers’ emotional well-being affects their dedication, innovation, decision-making, output, and tenure. Mood monitoring and management, therefore, are as crucial as mind management.
Skills	Better health and a positive outlook on life are also linked to EI. A person’s ability to identify and control emotions is greatly enhanced by high EI, which profoundly affects their relationships with others. They will better communicate with others and understand their requirements, emotions, and responses.

**Table 3 behavsci-12-00511-t003:** Demographic representation of Survey.

Demographic		
Gender	Male	43%
	Female	57%
Age	25–30 years	39%
	31–40 years	13.2%
	41–50 years	29.6%
	51–60 years	13.2%
	Above 60 years	5%
Occupation	Assistant Professor	63%
	Associate Professor	17%
	Professor	9%
	Lecturer	5%

**Table 4 behavsci-12-00511-t004:** Results of Reliability and Validity Analysis.

Variable	AV	CR	Items	R_sq	CV- Communality	CV- Redundancy
Attitude	0.547	0.789	4	0.742	0.089	0.392
EI	0.535	0.792	4		0.112	
Performance	0.688	0.912	4		0.56	
Knowledge	0.567	0.854	5	0.96	0.37	0.552
Skills	0.5	0.86	4	0.962	0.339	0.467

Source: The authors.

**Table 5 behavsci-12-00511-t005:** Results of Convergent Validity Analysis (Factor loading).

Constructs	Factors	Loadings
EI	E11	0.591
	E12	0.771
	E13	0.902
	E14	0.818
Performance	P1	0.870
	P2	0.82
	P3	0.389
	P4	0.590
Knowledge	K1	0.977
	K2	0.975
	K3	0.317
	K4	0.542
	K5	0.469
Skills	S1	0.918
	S2	0.532
	S3	0.466
	S4	0.485
Attitude	A1	0.800
	A2	0.701
	A3	0.869
	A4	0.893

Source: The authors.

## Data Availability

Not applicable.

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
