# Peer review of "The Relationship between Emotional Intelligence and Educators’ Performance in Higher Education Sector"

_behavsci, 2022, doi:10.3390/bs12120511_

Round 1
Reviewer 1 Report
First of all I would like to congratulate the authors on having this great idea of investigating the interlinkage of EQ and higher education! It is a nice endeavor and might be interesting for scientific teacher all over the world.
Still, I see some room for improvement in this paper:
First of all I find the title irritating, as you do not refer to the “normal” employee” but exclusively to scientific staff at Universities. As this is a relevant limitation, it should also be reflected in the title, and should be clearly depicted even in the initial section of the paper
Secondly, I miss the consistency throughout the theoretical parts. In the theoretical section you talk about “components of emotional intelligence are emotional perception, facilitating cognition, emotional understanding, and emotional control” (l. 84 /85). You explain that a bit, but not all of the concepts, then suddenly shift to stress process (and I do not know why, so how does the correlation between stress and emotion come about?) And then again you talk about trait-based and ability-based EI, where again I do not find the connection to the things mentioned before. Please rethink this section and try to provide a clear and consistent line of argument. But confusion continues a little later as in the research method (the correct term, not “methodology”, which is actually the science of devising methods) you bring into play three facets of EI: maturity, stability and competence (l.226). How does that fit with your theoretical section? Please lay the foundation for this categorization in the theoretical section! And then in your results you talk about emotional receptivity, maturity and competence (l.249) which have not been mentioned before. So, this is all very confusing and inconsistent. I would advise you to look at the paper as a whole and rethink and revise the complete theoretical background and foundation!
Thirdly, I would ask you to provide the empirical foundation for your evaluation of the emotional “qualities” of teacher (as provided on p. 6, bottom third). You describe things that seem to be a result of your survey, but there is no reference whatsoever. This needs to be amended.
Fourthly, it would be appropriate to argue why you chose this method and not maybe a qualitative approach, as you claim to open a new field. So, it might make sense to work on validity more than on representativity. But this is a side topic.
Furthermore, you might rethink the way you prepare the graphic presentation of date, e.g., the demographic description of your sample, which I personally find really complicated and not intuitive. Furthermore, the figuring in tables/charts is missing and the charts are not all referred to in the text either. This must be amended as well.
Finally, some minor amendments, like adding “real” research question rather at the beginning of the text. Some minor amendments in the language will also be necessary, I suggest asking a native reader to edit the text.
All in all, this is an interesting topic, but some serious amendment will have to be done to the scientific foundation before resubmitting the paper.
Author Response
Reviewer 1- Comments:
First of all I would like to congratulate the authors on having this great idea of investigating the interlinkage of EQ and higher education! It is a nice endeavor and might be interesting for scientific teacher all over the world.
Still, I see some room for improvement in this paper:
Comment 1: First of all I find the title irritating, as you do not refer to the “normal” employee” but exclusively to scientific staff at Universities. As this is a relevant limitation, it should also be reflected in the title, and should be clearly depicted even in the initial section of the paper
Response1 :
First of all, we are thankful to you for acknowledging our efforts. And, thank you for highlighting our areas of improvements. We understood your concerns and we have revised the title according to your instructions. The new title now is “The relationship between emotional intelligence and educa-tors' performance in higher education sector”.
Comment 2:
Secondly, I miss the consistency throughout the theoretical parts. In the theoretical section you talk about “components of emotional intelligence are emotional perception, facilitating cognition, emotional understanding, and emotional control” (l. 84 /85). You explain that a bit, but not all of the concepts, then suddenly shift to stress process (and I do not know why, so how does the correlation between stress and emotion come about?) And then again you talk about trait-based and ability-based EI, where again I do not find the connection to the things mentioned before. Please rethink this section and try to provide a clear and consistent line of argument. But confusion continues a little later as in the research method (the correct term, not “methodology”, which is actually the science of devising methods) you bring into play three facets of EI: maturity, stability and competence (l.226). How does that fit with your theoretical section? Please lay the foundation for this categorization in the theoretical section! And then in your results you talk about emotional receptivity, maturity and competence (l.249) which have not been mentioned before. So, this is all very confusing and inconsistent. I would advise you to look at the paper as a whole and rethink and revise the complete theoretical background and foundation!
Response 2:
As you Highlighted that consistency was missing in our paper. We have completely revised the study and added recent study that is consistent, we have also changed theoretical background and foundation and now it is completely relevant to the discussion and studies. For example, see page 3 and 4.
Comment 3:
Thirdly, I would ask you to provide the empirical foundation for your evaluation of the emotional “qualities” of teacher (as provided on p. 6, bottom third). You describe things that seem to be a result of your survey, but there is no reference whatsoever. This needs to be amended.
Response 3:
We have added the required references and amendments are made as highlighted by you.
Comment 4:
Fourthly, it would be appropriate to argue why you chose this method and not maybe a qualitative approach, as you claim to open a new field. So, it might make sense to work on validity more than on representativity. But this is a side topic.
Response 4:
Your comment is valid, we have explained this point in the research method Section, You can refer to 5 (We conducted a quantitative research to achieve greater information and understanding of the social world. We decided…..).
Comment 5:
Furthermore, you might rethink the way you prepare the graphic presentation of date, e.g., the demographic description of your sample, which I personally find really complicated and not intuitive. Furthermore, the figuring in tables/charts is missing and the charts are not all referred to in the text either. This must be amended as well.
Response 5:
Thanks for the great comment. We have modified all tables and referred to in the text. For example, see page 6-9.
Comment 6:
Finally, some minor amendments, like adding “real” research question rather at the beginning of the text. Some minor amendments in the language will also be necessary, I suggest asking a native reader to edit the text.
Response 6:
We referred our paper to a native editor and proofreader and he had made the required changes in the document and edited the text.
Comment 7:
All in all, this is an interesting topic, but some serious amendment will have to be done to the scientific foundation before resubmitting the paper.
Response 7:
We have made the necessary amendments and the paper has been modified accordingly according to your instructions. We hope you find it satisfactory this time.
Reviewer 2 Report
Dear Authors,
thank you for the opportunity! The research is great, although my advise is to further expand the study - making the theory and conclusion stronger! Find my comments in attachement.
All the best!

Author Response
Formatting issues: - Comments:
Comment 1:
Dear Authors!
I see that this work is a result of a hard work. There are some issues which you can find below.
These are the formatting issues which have to be corrected:
- “WILFRID LAURIER UNIVERSITY” – only the first letters should be with capital letter
Response 1:
“WILFRID LAURIER UNIVERSITY” has been written as “Wilfrid Laurier University”.
Comment 2:
- The first keyword (“Competences”) should be written with small first letter too
Response 2:
The first keyword (“Competences”) has be written with small letters
Comment 3:
- I think that Figure 1 should be smaller and not hanging into the margin at the left side
Response 3:
We have fixed the Margins of all the figures and tables now they all are within the margins.
Comment 4:
The “Variable description” – which is the name of the table – is on the previous page. Also, it is not indicated that it is a table. The tables are not numbered at all.
Response 4:
We have fixed the tables and now they are duly numbered.
Comment 5:
- The 2nd table – information about the sample – is also not clear. Why do you make several columns? Make it like this: Also, the category “41-50” is missing, and then the “50-60” should be changed to “51-60”.
Your Table 1 is Table 3. Check all of these.
Response 5:
Thanks for the great comment. Missing category is inserted along with its value, and column are also fixed.
Comment 6:
- Also, the formatting of your Table 2 should be changed. According to the template internal lines are not needed.
Response 6:
We have fixed the formatting of the table.
We appreciate your time and great/valid comments on the formatting! We tried our best to fix all issues according to your comments. We are also confident that the journal team will double check all of the formatting issue and fix them during the paper production stage. Thanks again
Content issues:
Comment 1:
- You wrote that: “Recent studies suggest the importance of emotional intelligence in the workplace”. You mention a study from 2017. I also advise you to use the study of “Mura et al. (2021)”. The study was issued at the end of 2021, so it is quite recent and also gained lot of citations through the year. I think you could find some more interesting thoughts in this article.
Response1:
Thanks for this great study! We used it based on your recommendation and it definitely improved the theoretical background section. Kindly, see page 3.
Comment 2:
- Using 58 sources is great, however the length of the theoretical part should be longer. For e.g. you should tell a more about why did you chose the Bar-on scale? For e.g. Mura et al (2021) discuss the scales, like LEAS, MSCEIT, Bar-On and AES. You should also tell more about these scales and justify your choice.
Response 2:
We completely agree with you, Now, we strengthen the paper and now the total references increased to 76 instead of 58.
Comment 3:
- Also, the Conclusions should be expanded as well.
Response 3:
we have explained the conclusion in more detail so the length of conclusion has expanded as required. Kindly see page 10.
Comment 4:
- I highly appreciate the inclusion of limitations and future directions at the end! However, these should be further expanded too!
Response 4:
We appreciate this comment. We expanded the limitations and future directions according to your comment.
The mentioned literature:
Mura, L., Zsigmond, T., & Machová, R. (2021). The effects of emotional intelligence and ethics of SME employees on knowledge sharing in Central-European countries. Oeconomia Copernicana, 12(4), 907–934. https://doi.org/10.24136/oc.2021.030
Round 2
Reviewer 1 Report
Dear Authors,
congrats, the paper is much better now, I really enjoyed reading it again. There is (or are possibly some) minor errors, e.g. one sentence comes twice (lines 100 to 105), also the abbreviation EI is mentioned twice (cf. l. 85 and 88). So you should read your paper thoroughly again, but after that I would see it fit for publication.
My congratulations!
Author Response
We are very happy that you enjoyed reading our paper. Thanks a lot for brining those minor errors to our attention. Fixing these errors will definitely strengthen the article. We have revised the paper according to your comments. In addition, we hope the journal team will highlight any similar issue(s) during the paper production stage.
Again, we really appreciate your time and efforts!